# Impact of the COVID-19 Pandemic on the Dutch Screening Program for Developmental Dysplasia of the Hip—Delayed Screening and One-Year Outcomes

**DOI:** 10.3390/children12050538

**Published:** 2025-04-23

**Authors:** Jan H. Hinloopen, Demi J. Donker, Joost H. van Linge, Christiaan J. A. van Bergen, Florens Q. M. P. van Douveren, Margret Foreman-van Drongelen, Frederike E. C. M. Mulder, Jaap J. Tolk, Pieter Bas de Witte

**Affiliations:** 1Department of Orthopaedics, Leiden University Medical Center, Albinusdreef 2, P.O. Box 9600, Postzone J-11-S, 2300 RC Leiden, The Netherlandsp.b.de_witte@lumc.nl (P.B.d.W.); 2Department of Orthopaedics, Reinier Haga Orthopedisch Centrum, Toneellaan 2, 2725 NA Zoetermeer, The Netherlands; 3Department of Orthopaedics, Amphia Ziekenhuis, Molengracht 21, 4818 CK Breda, The Netherlands; 4Department of Orthopedics and Sports Medicine, Erasmus MC–Sophia Kinderziekenhuis, Dr. Molewaterplein 60, 3015 GJ Rotterdam, The Netherlands; 5Department of Orthopaedic Surgery & Trauma, Máxima Medical Centre, P.O. Box 7777, 5500 MB Veldhoven, The Netherlands; 6Department of Hip Sonography, Diagnostiek voor U (Medical Diagnostic Center), Boschdijk 1119, 5626 AG Eindhoven, The Netherlands; 7Department of Orthopaedic Surgery, Care and Public Health Research Institute (CAPHRI), Maastricht University, 6200 MD Maastricht, The Netherlands

**Keywords:** DDH, COVID-19, developmental dysplasia of the hip, screening

## Abstract

Background/Objectives: In the Netherlands, selective ultrasound (US) screening for developmental dysplasia of the hip (DDH) typically occurs at 3 months of age. During the COVID-19 pandemic, US screening was temporarily halted in Dutch hospitals, with consequent delay in DDH screening and possibly inferior outcomes in DDH patients. Methods: We analyzed 1849 infants screened for DDH during the COVID-19 pandemic (March–August 2020) and 1663 infants screened before the pandemic (March–August 2019). We compared mean age and timing of screening (standard vs. delayed (delayed defined as ≥15 weeks)). For secondary outcomes, we compared DDH patients with delayed screening to standard screening, assessing severity at diagnosis, treatment method and duration, and outcomes at the age of one year, including acetabular index (AI) on radiographs. Results: Mean age at screening was 17.3 weeks during the COVID-19 crisis (2020) vs. 15.8 weeks in the 2019 cohort (mean difference 1.5, 95% CI 1.1–1.8, *p* < 0.001). Delayed screening occurred in 57.6% of infants in 2020 vs. 36.7% in 2019 (*p* < 0.001). Patients with DDH with delayed screening (*n* = 284), compared to standard screening (*n* = 284), did not differ in mean alpha angle at diagnosis (55.0° vs. 54.4°, mean difference 0.6, 95% CI −0.06–1.25, *p* = 0.08) and AI at one year (24.0° vs. 24.5°, mean difference −0.5, 95% CI −1.05–0.14, *p* = 0.13). Conclusions: This study revealed that disruption of healthcare caused by the COVID-19 pandemic resulted in a delay in the Dutch DDH-screening program. However, in this study, delayed screening was not associated with inferior outcomes at the age of one year.

## 1. Introduction

Developmental dysplasia of the hip (DDH) is a disorder affecting the developing proximal femur and acetabulum of the newborn [1,2]. Estimated incidence rates range between 0.1 and 6.6 per 1000 infants, depending on ethnicity and diagnostic method [3]. If diagnosed in the first twelve weeks of life, DDH can often be treated successfully with conservative measures [4,5]. However, if left untreated, DDH can lead to chronic pain, gait abnormalities, and early osteoarthritis of the hip [2,6]. Therefore, many countries have implemented screening programs to diagnose and treat DDH in early infancy.

The first months of the COVID-19 pandemic had a significant impact on global healthcare systems, resulting in impairment of various standard healthcare services [7,8,9]. In many countries, the screening program for DDH was also affected, causing a delay in DDH screening [10,11]. The precise magnitude and effect of this delay are not yet well understood. Therefore, we have evaluated the timing of the DDH screening in the Netherlands during and before the COVID-19 pandemic, as well as the consequences of delayed screening with regard to the outcomes of DDH at the age of one year.

In clinical practice in the Netherlands, all infants are screened at children’s healthcare centers at regular time points. At the age of 4 weeks, risk factors (RF) for the development of DDH (i.e., positive family history for DDH and/or hip osteoarthritis <50 years of age, breech position after 32 weeks of gestation and/or during delivery) are assessed, and physical examination (PE) of the hips is performed by a youth physician [12]. Another PE is performed by the youth physician at the age of three months. If any RF or abnormalities with PE are present at the 4-week or 3-month clinical screening, the infant is referred to a pediatric orthopedics department or medical diagnostic center for further examination and ultrasound (US) evaluation of the hips. According to the guidelines of the Dutch Orthopedic Association (NOV) and the Dutch Youth Healthcare Association (JGZ), this evaluation should take place at the age of 3 months, or earlier in case of a clinically suspected dislocation of the hip [12,13]. However, when abnormalities in PE are found at the 3-month clinical screening, children should be referred within 2 weeks. Therefore, children who are screened below the age of 15 weeks are screened in time according to the national guideline. Additionally, when abnormalities are found at a later age, children are referred to a pediatric orthopedics department for screening with US or radiographs.

The current literature provides no consensus on the optimal timing of screening for DDH, and the effects of a delay in screening and start of treatment are unclear (i.e., above the age of 3 months). Some studies have reported that late diagnosis is associated with an increase in surgical interventions and long-term complications [4,5]. Therefore, many countries perform US screening in the first 6 weeks of life [6,14]. However, early screening can lead to overdiagnosis and overtreatment because of a potential self-limiting course of immature hips or DDH before the age of 3 months [15,16]. On the other hand, it has been reported that screening at the age of 5 months is not associated with an increase in incidence rates and severity of DDH compared to screening at 3 months [17]. However, drawing conclusions with these data is difficult because it requires comparing heterogeneous diagnostics and treatment strategies in different populations in various countries. Hence, the optimal age to screen for DDH and initiate treatment remains unclear. The delay in DDH screening during the COVID-19 pandemic offers the opportunity to gain more insight on this subject by investigating the potential effects of a delay in diagnosis and treatment of DDH within a single country, patient population, and healthcare system.

The main objective of this multicenter retrospective study is to evaluate the delay in DDH screening in the Netherlands during the first wave of the COVID-19 crisis (March to August 2020), compared to a cohort from before the pandemic (March to August 2019). Secondary objectives are to assess whether patients diagnosed with DDH after delayed screening had a higher grade of dysplasia at the time of screening and/or inferior results with a 1-year follow-up compared to DDH patients with standard timing of screening.

## 2. Materials and Methods

The institutional ethics committee approved this retrospective multi-center study (protocol G20.120). Data from medical records of all infants who were screened using US or plain radiography between March and August 2020 (i.e., during the first COVID-19 wave; pandemic group) and between March and August 2019 (control group) were pseudonymously collected and analyzed. All patients under the age of one year referred to the participating centers for first-time DDH screening were eligible for inclusion in the study. Exclusion criteria were parents using an opt-out option for using their children’s data for research and children who presented with secondary hip dysplasia because of neuromuscular disorders or congenital syndromes.

After referral by a youth healthcare physician or general practitioner, the infants were screened for DDH by US or plain radiography (generally when aged > 6 months and/or an inconclusive US due to the ossific nucleus). Patients were then examined by a pediatric orthopedic surgeon or by a supervised orthopedic resident. To limit physical contact between the physicians, patients, and parents during the first COVID-19 lockdown, PE of the hips was only performed when the US or radiograph showed (borderline) abnormalities or when the reason for referral was abnormal PE.

Ultrasounds and radiographs were performed according to usual care by trained and experienced radiologists or ultrasonographers. The orthopedic surgeon and/or radiologist then evaluated the imaging results. For US evaluation, the Graf classification system was applied by measuring the alpha and beta angles and assessing lateralization of the femoral head [18]. Evaluation of the radiographs was performed by measuring the acetabular index (AI) and assessing the (sub)luxation of the femoral head [19].

Patients with DDH were treated according to the Dutch Guideline with either watchful waiting or abduction splint treatment as the first step, depending on the severity of the condition and the age of the patient [13,20,21]. Mild cases of DDH (Graf types IIa, IIb, and IIc) underwent follow-up US every 6 weeks. In case of progression at 6 weeks after diagnosis or no normalization after 12 weeks of watchful waiting, treatment was initiated using an abduction splint (Pavlik harness (or CAMP abduction brace in specific cases)). In case of a (sub)luxated hip (Graf D, III, or IV), treatment was started with a Pavlik harness, with US follow-up at 3–4 and 6–8 weeks. In case of no centralization, closed reduction was performed under general anesthesia, followed by an abduction spica cast for the duration of 3 months. For this study, the clinical follow-up data of the children with DDH were included up until the age of 1 year.

The following baseline characteristics were evaluated: demographics, prematurity, RF, PE findings (normal, abduction limitation, abduction difference, Galeazzi, Barlow, Ortolani, combination, not performed), indication for screening, and type of screening. For continuous data, data distributions were evaluated with histograms. Normally distributed data were presented with means and standard deviations, and skewed data with medians and ranges. Categorical data were presented with counts and percentages.

For the primary outcome, mean age at the time of initial screening (US or radiographs) was compared between the pandemic group and the control group using the student’s *t*-test. The age was adjusted in case of prematurity, in compliance with the Dutch guidelines [12]. Furthermore, the proportion of children who underwent delayed screening (defined as screening above the age of >15 weeks) was compared between the two groups using the χ^2^-test. The distribution of screened infants per month was also evaluated and compared between both periods using the χ^2^-test.

For the secondary outcomes, we compared all DDH patients with delayed screening (i.e., from both study periods) with all DDH patients who had standard timing of screening. We assessed the same baseline characteristics for these groups. A comparison of the initial imaging data was performed: alpha angles and Graf classification for US, and AI and (sub)luxation for radiographs of all affected hips. For the comparison of alpha angles, we only used the values of Graf IIa, IIb, IIc, and D hips because measurement of alpha angles in Graf III and IV hips are usually unreliable due to the decentered position of the femoral head [18]. Mean alpha angles were compared using Student’s *t*-test. Proportions of patients with severe DDH (Graf D or higher) were compared between both groups using the χ^2^-test. For outcomes at the age of one year, we compared mean AI using Student’s *t*-test and (sub)luxation with Fisher’s exact test.

Additionally, treatment outcome was compared in terms of the performed type(s) of treatment, including watchful waiting, Pavlik harness, CAMP abduction brace, or surgery (closed and/or open reduction). The proportion of children who underwent surgery was compared between the two groups using the χ^2^-test. Treatment duration was measured for Pavlik harness, CAMP abduction splint, and spica cast (after open or closed reduction). Duration of Pavlik harness and CAMP abduction splint was divided into three groups (6 weeks, 12 weeks, and >12 weeks), and duration of spica cast was also divided into three groups (2 months, 3 months, and >3 months). The proportions of the treatment type duration were compared between both groups using χ^2^-tests. Finally, we compared proportions of complications (avascular necrosis of the femoral head (AVN), femoral nerve palsy, skin injury, bleeding, infection, and re-dislocation) between both groups.

Statistical analyses were performed in SPSS version 25. Differences were considered statistically significant if *p* < 0.05.

## 3. Results

During the 2020 pandemic period, 1683 infants were screened at the six participating hospitals and one diagnostic medical center, and 1867 in the 2019 control period. Thirty-eight infants were excluded because they did not fit the inclusion criteria. This resulted in, respectively, 1663 and 1849 infants available for analysis. Both groups were found to be comparable at baseline for demographics and RF except for gender and family history of DDH and PE (Table 1).

### 3.1. Primary Outcomes

With a mean age of 17.3 weeks, infants screened in the pandemic group were on average 1.5 weeks older at the time of screening than those in the control group (95% CI 1.1–1.8, *p* < 0.001) (Table 2). In the pandemic group, 57.6% (1065/1849) of infants had delayed screening, compared to 36.7% (610/1663) in the control group (odds ratio: 2.3, *p* < 0.001) (Table 2). The monthly distribution showed a significant decrease in screened children in March and April 2020 compared to March and April 2019 (*p* < 0.001 and *p* < 0.001) (Figure 1). In contrast, the number of screened infants from May to August 2020 was larger compared to the same months in 2019.

### 3.2. Secondary Outcomes

Overall, in both cohorts, there were 1837 infants with standard screening and 1675 with delayed screening. DDH was present in 17.0% (284/1675) of infants with delayed screening, compared to 15.5% (284/1837) of infants with standard screening (*p* = 0.23). The baseline characteristics of the two DDH patient groups can be found in Table 3. The mean age at screening in the standard group was 13.1 weeks, compared to 20.1 weeks in the delayed group (Table 3). Due to 179 infants with bilateral DDH, there were 359 affected hips in the delayed screening group and 388 in the standard screening group. In these DDH patients, mean alpha angle at the time of diagnosis was 55.0° in the delayed screening group vs. 54.4° for standard screening (mean difference 0.6, 95% CI: 0.1–1.3, *p* = 0.08) (Table 4). We found a significant difference in Graf classification distribution between both groups (*p* < 0.001) (Table 4). Severe DDH was present in 13.4% (48/359) of affected hips in the delayed screening group vs. 16.8% (65/388) in the standard screening group (odds ratio: 0.77, *p* = 0.20) (Table 4). Mean AI was 29.8° in the delayed screening group vs. 35.6° for standard screening (Table 5). In the delayed screening group, (sub)luxation was present in 25.0% (6/24) of affected hips vs. 20.0% (one out of five) in the standard screening group (Table 5).

At the one-year follow-up of DDH patients, the mean AI of the affected hips was 24.0° in the delayed screening group vs. 24.5° in the standard group (mean difference −0.5, 95% CI −1.1–0.1, *p* = 0.13) (Table 6). There was 1 dislocation on the radiographs at the age of one year in the standard screening group vs. 0 in the delayed screening group (*p* = 0.32) (Table 6). Total loss to the follow-up was 11.9% in the standard screening group and 8.4% in the delayed screening group.

See Table 7 for the types of treatment performed in both groups. We found no significant difference in the rate for surgical treatment between both groups, 6% (17/284) in the standard screening group vs. 4.2% (12/284) in the delayed screening group (*p* = 0.35). In most cases, treatment started with watchful waiting. If there was persistent dysplasia after 6 weeks, treatment using a Pavlik or CAMP was performed. In the standard screening group, 47 infants who started with watchful waiting needed treatment with an abduction device, vs. 40 infants in the delayed screening group. For treatment duration with an abduction device, we found no significant difference in the duration of Pavlik, CAMP, and spica cast treatment (Table 8).

The total number of complications was four in the delayed screening group and three in the standard screening group. In the delayed group, three infants had femoral nerve palsy, and one infant had AVN. In the standard group, two infants had femoral nerve palsy, and one infant had a redislocation.

## 4. Discussion

In this study, we evaluated the impact of the COVID-19 crisis on the Dutch DDH screening program in 3512 infants in seven centers. We found a significantly increased age at the time of screening for infants who were screened during the pandemic period of March to August in 2020, compared to the same period in 2019. Furthermore, there were substantially more infants with delayed screening (>15 weeks) in the pandemic period than in the control period: 57.6% vs. 36.7%. However, we found no longer duration of treatment or inferior outcomes at 1 year in children with delayed screening.

Several studies have examined the timing of DDH screening during the COVID-19 pandemic and found similar results to our study. Mert Doğan et al. reported a 5.5% increase in children who underwent delayed US screening (>3 months) in their study, comparing screening from April to July 2020 to the same period in 2019 in a Turkish hospital [22]. In a study by Guindani et al., the rate of delayed US screening in northern Italy during their lockdown period from 10 March 2020 to 3 May 2020 was 74% [11]. However, there were no comparisons made with delayed screening rates outside the COVID-19 pandemic. On the contrary, Ayaz et al. found no increase in delayed US screening (i.e., >3 months) in their study in Turkey [8]. The authors reported the annual rates of delayed screening from 2018 to 2021, which were the lowest in 2020 (5%), although this was not statistically significant. Differences in the results of these studies are likely caused by the differences in lockdown measures and available resources for DDH screening between countries and hospitals. In several of our participating hospitals, screening was altered to make sure children with abnormal PE were screened with priority. The added value of the present study is the investigation of the potential consequences of the delay in screening, diagnosis, and treatment in a large and multicenter cohort.

With regard to our reported monthly distributions of DDH screening, the study by Mert Doğan et al. showed similarities to our study [22]. Ultrasound screening decreased by 63% from April to May 2020, during the beginning of the pandemic, and increased by 94% from June to July, compared to the same months in 2019. In contrast, Ayaz et al. reported a decrease in screened children from March 2020 up until the end of the year, with no full recovery during the summer and autumn of 2020 [10]. This discrepancy could be caused by a difference in the pandemic restriction policies or the willingness of patients to visit the hospitals during the pandemic. In our study, the increase in patients in the summer months was caused by our hospitals also screening the delayed patients from the first months of COVID-19, in addition to the normal monthly number of patients.

When comparing our standard and delayed screening groups of DDH patients at baseline, we found that in the delayed group fewer children had RF 183 (64.4%) vs. 238 (83.8%) (*p* < 0.001) and more children had an abnormal PE 79 (27.8%) vs. 34 (12.0%) (*p* < 0.001). A significant number of infants will be referred solely based on the PE findings. If abnormalities are found at later standardized appointments (i.e., 3 and 5–6 months), the infants will be screened after the age of three months. This is why more infants with abnormal PE had delayed screening in our study. No significant differences in alpha angles and DDH severity were found at diagnosis between the delayed and standard screening groups. A difference in Graf classification was found because in the standard screening group there were 35 children with Graf IIa hips, and there were no children with Graf IIa hips in the delayed screening group, since these children are all 3 months or older. This was partly compensated by a higher number of Graf 2B hips in the delayed group. Additionally, this might support previous findings that premature hips or mild dysplastic hips follow a self-limiting course [15,16].

We found a mean difference of 7 weeks between our standard and delayed screening groups of DDH patients (13.1 weeks vs. 20.1 weeks). This difference resulted in no significant differences in AI-index and (sub)luxation on radiographs at the age of one year. These results are in line with the study of Geertsema et al. Comparing mean AI at the age of 18 months between DDH patients who were screened at the age of 5 months vs. 3 months, the authors found no significant differences between these two screening groups [17]. However, Phelan et al. found a higher rate of persistent dysplasia that required surgery in infants screened at the age of 3 months or older compared to those younger than 3 months [4]. These differences in results and preferred timing of screening are also found in the systematic review of Shorter et al. [6]. They concluded that there is not enough substantial evidence for recommendations for an optimal timing of screening.

No relevant differences in treatment types during the first year of life were found between the two groups. There were 12 (4.2%) infants who ultimately required closed reduction in the delayed screening group, compared to 17 (6.0%) infants in the standard screening group, which was not statistically significant. These findings are supported by the results of a recent systematic review conducted by Cheok et al., in which the prevalence of late diagnosis (age >3 months), abduction splint treatment, and surgical treatment was compared between universal screening and selective screening [23]. The authors found an increase in late diagnosis in the selective screening group vs. the universal screening group (0.45 vs. 0.10 per 1000 live births); however, this did not result in more surgical procedures: 0.49 vs. 0.48 per 1000 live births. Additionally, more infants received abduction splint treatment in the universal screening group (55.54 vs. 0.48 per 1000 live births). These results suggest that universal screening causes overtreatment in infants without reducing the rates of operative treatment.

We found no significant difference in the duration of Pavlik, CAMP, and spica cast treatment. A possible explanation for this finding could be that there were no differences in severity of DDH between both groups. However, the study by Bialik et al. suggests that treatment duration of Pavlik increases significantly in Graf IIc hips when started at the age of 13 weeks or older, compared to starting at the ages of 5 weeks, 6–8 weeks, and 9–12 weeks [24]. The authors also found the same results for Graf D, III, and IV hips; however, this was only significant when comparing 13 weeks or older with 5 weeks. The results of Bialik et al. primarily show a difference in duration between treatment onset below the age of 5 weeks and above the age of 13 weeks. In our study, most infants are older than 5 weeks, which could explain the discrepancy between the results.

We found no relevant differences in complications in DDH patients between the standard and delayed screening groups. Rosendahl et al. reported similar results in their randomized controlled trial [25,26]. In this study, the authors compared two treatment groups of children with stable DDH diagnosed in the first week of life, one group, which received direct treatment immediately after diagnosis (n = 64) and one group, which received active surveillance with US every 6 weeks (n = 64), resulting in 12 infants who received treatment at the age of three months and 5 infants at the age of six months. The authors found no signs of AVN in both groups after one and six years of follow-up.

A limitation of our study is the relatively small patient population. Despite our sample size of 3519 screened infants and 568 infants with DDH, this population might be insufficient to detect relevant clinical differences in invasive treatment and complication rates, as these outcomes are quite rare. For instance, AVN has reported rates of 0.7% to 16% in infants treated with a Pavlik harness or with open or closed reduction [27,28,29]. Furthermore, the duration of follow-up in this study is too short for an adequate detection of some of the AVN hips, since lower grades of AVN can cause deformities of the femoral head and neck, which typically present after the age of two years [30]. Additionally, the credibility of our results can be reduced due to the presence of multiple observers of the US images in some cases and due to a high reported interobserver variability in AI measurements [31,32], which, on the other hand, increases the external validity of our results. Finally, the retrospective nature of the study meant that the results rely on consistent medical records, which were used for the data collection.

## 5. Conclusions

Due to the COVID-19 crisis and the subsequent reduction in capacity of the DDH screening program in our hospitals, infants were screened at an older age, and a significant proportion underwent delayed screening, which shows that the hospitals could not completely comply with the Dutch guidelines on DDH screening. However, we found no clinically relevant differences at the age of one year in terms of persistent dysplasia, treatment, and complications, between the delayed and standard screening groups. These findings add to the ongoing discussion on the optimal timing of screening. Further research into the long-term follow-up of infants with delayed screening with a large sample size is needed to gain more insight into the effects of delayed diagnosis and treatment on more rare complications and outcomes. Agreements can be made between hospitals and local children’s healthcare centers to make sure children with abnormal PE can be screened with priority in future major disruptions of healthcare.

## Figures and Tables

**Figure 1 children-12-00538-f001:**
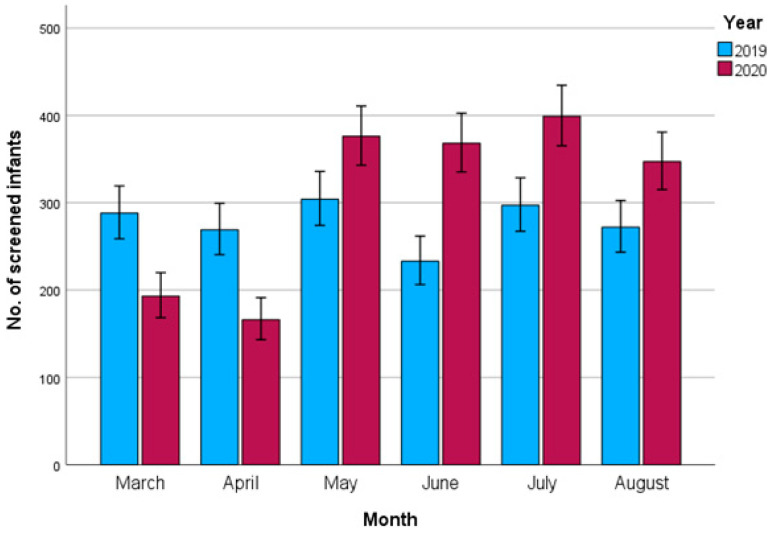
Monthly distribution of screened infants in 2019 (control) and 2020 (COVID-19 pandemic), showing a clear dip at the start of the COVID-19 pandemic, followed by higher numbers (i.e., catch-up growth) in May–August.

**Table 1 children-12-00538-t001:** Baseline characteristics at DDH screening.

	Control Group (2019) (n = 1663)	Pandemic Group (2020) (n = 1849)	*p*-Value
Gender (%)			
Female	951 (57.2)	941 (50.9)	**<0.001**
Premature (%)	103 (6.2)	106 (5.7)	0.56
Risk factors (%)			
None	239 (14.1)	223 (12.1)	**0.04**
Family history of DDH	439 (26.4)	567 (30.7)	**0.01**
Family history of hip OA ^a^	8 (0.5)	9 (0.5)	0.98
Breech	589 (35.4)	641 (34.7)	0.64
Clubfoot	25 (1.5)	19 (1.0)	0.21
Twin	24 (1.4)	27 (1.5)	0.97
Unsafe swaddle	4 (0.2)	1 (0.1)	0.20
Combination ^b^	123 (7.4)	154 (8.3)	0.31
Unknown	212 (12.7)	208 (11.2)	0.17
Physical examination (%)			
Normal	895 (53.8)	818 (44.2)	**<0.001**
Abduction limitation	94 (5.6)	124 (6.7)	0.17
Abduction difference	7 (0.4)	6 (0.3)	0.64
Galeazzi positive	32 (1.9)	33 (1.8)	0.76
Barlow positive	1 (0.1)	0 (0.0)	0.47
Ortolani positive	1 (0.1)	0 (0.0)	0.47
Combination ^c^	52 (3.1)	39 (2.1)	0.06
Not performed	582 (35.0)	829 (44.8)	**<0.001**
Indication for screening (%)			
Clinical suspicion	211 (12.7)	207 (11.2)	0.17
Positive risk factors	1149 (69.1)	1328 (71.8)	0.08
Both	59 (3.5)	79 (4.3)	0.27
Unknown	244 (14.6)	235 (12.7)	0.10
Type of screening (%)			
Ultrasound	1630 (98.0)	1805 (97.6)	0.42
Radiographs	21 (1.3)	34 (1.8)	0.17
Both	10 (0.6)	7 (0.4)	0.34

^a^ Family history of hip osteoarthritis < 50 years. ^b^ Combination of multiple risk factors. ^c^ Combination of multiple abnormal findings in physical examination. DDH, developmental dysplasia of the hip; OA, osteoarthritis. Bold text: statistically significant result.

**Table 2 children-12-00538-t002:** Mean age and timing of screening.

	Control Group (2019) (n = 1663)	Pandemic Group (2020) (n = 1849)	Mean Difference	95% CI	*p*-Value
Mean age at screening (±SD) ^a^	15.8 (±5.25)	17.3 (±5.51)	1.5	1.1–1.8	**<0.001**
Timing of screening (%)			N.A.	N.A.	**<0.001**
Standard	1053 (63.3)	784 (42.4)			
Delayed (≥15 weeks)	610 (36.7)	1065 (57.6)			

^a^ Mean age in weeks. SD, standard deviation; CI, confidence interval. N.A.: Not Applicable. Bold text: statistically significant result.

**Table 3 children-12-00538-t003:** Baseline characteristics of DDH patients: standard vs. delayed screening (≥15 weeks).

	Standard Screening (n = 284)	Delayed Screening (n = 284)	*p*-Value
Age (± SD)	13.09 (±1.66)	20.08 (±5.75)	**<0.001**
Gender (%)			
Female	206 (72.5)	204 (71.8)	0.85
Premature (%)	25 (8.8)	12 (4.2)	**0.03**
Risk factors (%)			
None	46 (16.2)	101 (35.6)	**<0.001**
Family history of DDH	90 (31.7)	70 (24.6)	0.06
Family history of hip OA ^a^	2 (0.7)	1 (0.4)	0.50
Breech	112 (39.4)	91 (32.0)	0.07
Clubfoot	0 (0.0)	0 (0.0)	N.A.
Twin	3 (1.1)	3 (1.1)	0.66
Unsafe swaddle	1 (0.4)	4 (1.4)	0.37
Combination ^b^	30 (10.6)	12 (4.2)	**0.01**
Unknown	0 (0.0)	2 (0.7)	0.50
Physical examination (%)			
Normal	202 (71.1)	185 (65.1)	0.13
Abduction limitation	31 (10.9)	54 (19.0)	**0.01**
Abduction difference	2 (0.7)	6 (2.1)	0.29
Galeazzi positive	10 (3.5)	8 (2.8)	0.63
Barlow positive	0 (0.0)	1 (0.4)	0.50
Ortolani positive	0 (0.0)	0 (0.0)	N.A.
Combination ^c^	30 (10.6)	22 (7.7)	0.24
Not performed	9 (3.2)	8 (2.8)	0.81
Indication for screening (%)			
Clinical suspicion	34 (12.0)	79 (27.8)	**<0.001**
Positive risk factors	201 (70.8)	149 (52.5)	**<0.001**
Both	33 (11.6)	23 (8.1)	0.16
Unknown	16 (5.7)	33 (11.7)	**0.02**
Type of screening (%)			
Ultrasound	277 (97.5)	263 (92.6)	**0.01**
Radiograph	0 (0.0)	17 (6.0)	**<0.001**
Both	5 (1.8)	3 (1.1)	0.48

^a^ Family history of hip osteoarthritis < 50 years. ^b^ Combination of risk factors. ^c^ Combination of abnormal findings in physical examination. DDH, developmental dysplasia of the hip; OA, osteoarthritis. N.A.: Not Applicable. Bold text: statistically significant result.

**Table 4 children-12-00538-t004:** Mean α angle and Graf classification in DDH population at time of screening.

	Standard Screening (n = 388) ^a^	Delayed Screening (n = 359) ^a^	Difference	*p*-Value
Mean α angle (±SD) ^b^	54.4 (±4.76)	55.0 (±4.05)	0.6 (95% CI 0.1–1.3) ^c^	0.08
Graf classification (%)			N.A.	**<0.001**
IIa	35 (9.0)	0 (0.0)		
IIb	260 (67.0)	284 (80.6)		
IIc	28 (7.2)	22 (6.1)		
D	36 (9.3)	31 (8.6)		
III	23 (5.9)	12 (3.3)		
IV	6 (1.3)	5 (1.4)		
Severe DDH (%) ^d^	65 (16.8)	48 (13.4)	0.77 ^e^	0.20

^a^ Number of affected hips. ^b^ Mean α angle in degrees. Only α angles of Graf IIa, IIb, IIc, and D. ^c^ Mean difference (95% CI). ^d^ Severe DDH defined as Graf type D, III, or IV. ^e^ Odds ratio. US, ultrasound; DDH, developmental dysplasia of the hip; SD, standard deviation; CI, confidence interval. Bold text: statistically significant result.

**Table 5 children-12-00538-t005:** Mean AI and (sub)luxation in DDH population at time of screening.

	Standard Screening (n = 5) ^a^	Delayed Screening (n = 24) ^a^
Mean AI (±SD)	35.6 (±9.5)	29.8 (±4.9)
(sub)luxation (%)	1 (20.0)	6 (25.0)

^a^ Number of affected hips. DDH, developmental dysplasia of the hip; AI, acetabular index.

**Table 6 children-12-00538-t006:** Radiological characteristics in DDH population at age of 1 year.

	Standard Screening (n = 342) ^a^	Delayed Screening (n = 329) ^a^	Mean Difference	95% CI	*p*-Value
Mean AI (±SD) ^b^	24.5 (±4.05)	24.0 (±3.37)	−0.5	−1.1–0.14	0.13
(Sub)luxation (%)	1 (0.3)	0 (0.0)	N.A.	N.A.	0.32

^a^ Number of affected hips. ^b^ Mean AI in degrees. DDH, developmental dysplasia of the hip; AI, acetabular index; SD, standard deviation; CI, confidence interval. N.A.: Not Applicable.

**Table 7 children-12-00538-t007:** Performed treatment in DDH population.

	Standard Screening (n = 284) ^a^		Delayed Screening (n = 284) ^a^	
	Graf IIa/b/c	Graf D/III/IV	Graf IIa/b/c	Graf D/III/IV
Watchful waiting (%)	183 (64.4)	0.0 (0.0)	156 (54.9)	0 (0.0)
Pavlik harness (%)	55 (19.4)	57 (20.1)	86 (30.3)	40 (14.1)
CAMP abduction brace (%)	3 (1.1)	16 (5.6)	18 (6.3)	12 (4.2)
CR without tenotomy (%)	0 (0.0)	13 (4.6)	1 (0.4)	6 (2.1)
CR with tenotomy (%)	0 (0.0)	4 (1.4)	0 (0.0)	5 (1.8)
OR (%)	0 (0.0)	0 (0.0)	0 (0.0)	0 (0.0)

^a^ Number of infants. DDH, developmental dysplasia of the hip; CR, closed reduction; OR, open reduction.

**Table 8 children-12-00538-t008:** Treatment duration in DDH population.

	Standard Screening (n = 148) ^a^	Delayed Screening (n = 168) ^a^	*p*-Value
Pavlik total (%)	112	126	0.62
6 weeks	53 (47.3)	66 (52.4)	
12 weeks	51 (45.5)	54 (42.9)	
>12 weeks	8 (7.2)	6 (4.7)	
CAMP total (%)	19	30	0.94
6 weeks	1 (5.3)	1 (3.3)	
12 weeks	15 (78.9)	24 (80.0)	
>12 weeks	3 (15.8)	5 (16.7)	
Spica cast total (%)	17	12	0.75
2 months	0 (0.0)	0 (0.0)	
3 months	14 (82.4)	11 (91.7)	
>3 months	3 (17.6)	1 (8.3)	

^a^ Number of infants. DDH, developmental dysplasia of the hip.

## Data Availability

The data presented in this study are available from the corresponding author upon request.

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
