# Peer review of "Impact of the COVID-19 Pandemic on the Dutch Screening Program for Developmental Dysplasia of the Hip—Delayed Screening and One-Year Outcomes"

_children, 2025, doi:10.3390/children12050538_

Round 1

Reviewer 1 Report

Comments and Suggestions for Authors

General comments

This manuscript analyses the impact of the COVID-19 pandemic on the screening program for developmental dysplasia of the hip (DDH) in the Netherlands, focusing on the outcomes of delayed screenings. This manuscript aims 1 to evaluate the delay in DDH screening in the Netherlands during the first wave of the COVID-19 crisis, compared with a cohort from before the pandemic, and 2 to assess whether patients diagnosed with DDH after delayed screening had a higher grade of dysplasia at the time of screening and/or inferior results with 1-year follow-up, compared with DDH patients with standard timing of screening. The authors’ aims are commendable. The authors found that the mean age at screening during the COVID-19 pandemic increased to 17.3 weeks compared with 15.8 weeks before the pandemic, with delayed screening occurring more frequently. However, they observed no significant differences in diagnostic severity or one-year outcomes—such as alpha angle or acetabular index—between delayed and standard screening groups. Overall, the authors manage to fulfil their aims sufficiently.

Minor comments

(line 32 and elsewhere throughout the manuscript, as well) Please, do not start sentences with acronyms;

Please, add some details about statistical analysis;

Table 3 and Figure 1 not introduced in-text;

Ref 23 should be cited “earlier”.

Author Response

Thank you very much for taking the time to review this manuscript. Please find the detailed responses below and the corresponding revisions/corrections highlighted in track changes in the re-submitted files.

General comments

This manuscript analyses the impact of the COVID-19 pandemic on the screening program for developmental dysplasia of the hip (DDH) in the Netherlands, focusing on the outcomes of delayed screenings. This manuscript aims 1 to evaluate the delay in DDH screening in the Netherlands during the first wave of the COVID-19 crisis, compared with a cohort from before the pandemic, and 2 to assess whether patients diagnosed with DDH after delayed screening had a higher grade of dysplasia at the time of screening and/or inferior results with 1-year follow-up, compared with DDH patients with standard timing of screening.

The authors’ aims are commendable.

Thank you for your compliment. We hope our article indeed gives insights in 1) the overall performance of the Dutch selective screening program, 2) delays during the covid period, 3) the effects of delayed screening on treatment outcome.

The authors found that the mean age at screening during the COVID-19 pandemic increased to 17.3 weeks compared with 15.8 weeks before the pandemic, with delayed screening occurring more frequently. However, they observed no significant differences in diagnostic severity or one-year outcomes—such as alpha angle or acetabular index—between delayed and standard screening groups. Overall, the authors manage to fulfil their aims sufficiently.

Thank you for pointing this out. Although we did find significantly delayed screening during the COVID period, the significance of this finding is not clear. We did not find significant differences between the outcomes of patients with standard and delayed screening.

Another remarkable finding was, that even in the period before the COVID crisis, there was delayed screening in quite some patients. This finding has led to discussions within our national pediatric orthopedics foundation, stressing our members to ensure proper screening at 3 months.

Minor comments

(line 32 and elsewhere throughout the manuscript, as well) Please, do not start sentences with acronyms;

Thanks. We addressed this throughout the manuscript, and marked these changes.

Please, add some details about statistical analysis;

Our statistics section was written concise and point-by-point. We have now elaborated somewhat more on our statistical approach (lines 149-163), as well as the comparisons between the standard and delayed groups. Please feel free to comment if there are remaining unclarities.

Table 3 and Figure 1 not introduced in-text;

We have now referred to these in the manuscript (lines 185-186 and lines 178-180)

Ref 23 should be cited “earlier”.

We have resolved this issue.

Reviewer 2 Report

Comments and Suggestions for Authors

This paper describes a study of 1849 vs. 1663 patients screened for DDH and compares the outcomes of this during the covid pandemic and prior to it, with 1 year follow-up. It is a large dataset and I would recommend it for publication. The paper provides an overall reassuring message that DDH care was not compromised in this population group during the covid pandemic. Please see below for comments and suggested edits:

Introduction

A good introduction is provided with background on the topic. Please however provide more clarity on “early stage” (line 46) with specific dates i.e. < 12 weeks as per your references.

Please specify in the text why you chose 15 weeks as the cut off for a delayed screening. Is this based on a clinical basis or was this chosen as the best point to separate your two groups? You state in line 66 that US should be peformed by age 3 months, which would suggest 12 weeks a more appropriate cut off than 15 weeks. Can you advise why you chose 15 weeks instead of 12 weeks as the cut off and provide this explanation in the paper to contextualise and help the readers understanding. It may benefit your paper to provide a subgroup analysis with different time points as the cut off, and how many patients were considered “delayed” on this. i.e. a cut off of 6 weeks (as per line 74), cut off of 12 weeks (as per line 72) and your chosen cut off of 15 weeks. You can then show that at each stage of delay, there was no negative outcomes at 1 year (and therefore focus your argument more on the assumption that a US date of 15 weeks is non-inferior to 12 or 6 weeks).

Methods

Appropriate ethical approval stated. Does your data display as parametric data? Age at the time of an investigation would likely present as non-parametric data and should be displayed as median values (not mean) with range. For example, the 2019 data displays 15.8 as the mean value, with 15 weeks highlighted as the cut-off for delayed screening, but only 36.7% of patients subsequently reported as delayed – this implies large outliers which are bringing the mean value up and distorting your results, I suspect the use of median would be more appropriate and likely improve the message of your paper. It is not as strong an impact showing a difference of only 1.5 weeks between groups, when there is a larger difference of 57.6% vs 36.7% delayed

A separate analysis correlating time of diagnosis (as a range) to outcomes would be beneficial, rather than only delineating into two groups only

Results

Good presentation of results with good use of tables

Discussion

I think the message of your discussion and conclusion should be altered, It ambiguously implies that a delay in screening is not associated with inferior outcomes. It should be highlighted that the delay is only 1.5 weeks (17.3 vs. 15.8 weeks). This delay should also be discussed in the context of the wider literature – what is already known about the optimum age for diagnosis and at what point a delay in diagnosis is associated with worsened outcomes. Based on this you should highlight to the reader what new information your research brings to the wider research base. Further discussion on the topics raised in the methods comments is recommended.

Author Response

Thank you very much for taking the time to review this manuscript. Please find the detailed responses below and the corresponding revisions/corrections highlighted in track changes in the re-submitted files.

Introduction

A good introduction is provided with background on the topic. Please however provide more clarity on “early stage” (line 46) with specific dates i.e. < 12 weeks as per your references.

Thank you for this compliment. We have defined “early stage” in the manuscript.

Please specify in the text why you chose 15 weeks as the cut off for a delayed screening. Is this based on a clinical basis or was this chosen as the best point to separate your two groups? You state in line 66 that US should be peformed by age 3 months, which would suggest 12 weeks a more appropriate cut off than 15 weeks. Can you advise why you chose 15 weeks instead of 12 weeks as the cut off and provide this explanation in the paper to contextualise and help the readers understanding.

We have now clarified this more in the manuscript (lines 66-72) . We did not do separate analyses for different cut-offs. The 15 weeks was chosen on practical grounds: according to the Dutch guideline, children should be referred for ultrasound screening at 3 months (i.e. 13 weeks) in children with risk factors. In practice ultrasounds planning might variate 1-2 weeks around this 13-week moment.

Furthermore, when abnormalities are found with physical examination by the youth physician at 3 months, infants must be referred for ultrasound screening within 2 weeks  (i.e. at the age of around 15 weeks).

It may benefit your paper to provide a subgroup analysis with different time points as the cut off, and how many patients were considered “delayed” on this. i.e. a cut off of 6 weeks (as per line 74), cut off of 12 weeks (as per line 72) and your chosen cut off of 15 weeks. You can then show that at each stage of delay, there was no negative outcomes at 1 year (and therefore focus your argument more on the assumption that a US date of 15 weeks is non-inferior to 12 or 6 weeks).

Thank you for this suggestion. This would not really change the analyses of our primary research question (i.e. mean age at screening in both time periods), but it would change the analyses for the secondary outcomes (changes in outcome between “standard” and “delayed” screening).

In our opinion, adding sensitivity analyses with 2-3 additional groups would lead to a lot of additional outcomes and tables in our manuscript, making it harder to read and interpret for the reader.

However, if desired by the reviewer(s) and/or editor, we would of course be willing to add analyses with a different cut-off point (e.g. 6 and/or 12 weeks) in a supplement.

Methods

Appropriate ethical approval stated.

Does your data display as parametric data? Age at the time of an investigation would likely present as non-parametric data and should be displayed as median values (not mean) with range. For example, the 2019 data displays 15.8 as the mean value, with 15 weeks highlighted as the cut-off for delayed screening, but only 36.7% of patients subsequently reported as delayed – this implies large outliers which are bringing the mean value up and distorting your results, I suspect the use of median would be more appropriate and likely improve the message of your paper.

Thank you for this comment. We checked these data for normality, which we tried to explain in the Methods section (lines 131-132). We therefore chose to use parametric tests.

It is not as strong an impact showing a difference of only 1.5 weeks between groups, when there is a larger difference of 57.6% vs 36.7% delayed

We wanted to display the mean difference, as well as the difference bases on a (debatable) cut-off point. Therefore, we also mention the mean difference, which also gives more insight in the magnitude of the difference between both groups.

A separate analysis correlating time of diagnosis (as a range) to outcomes would be beneficial, rather than only delineating into two groups only

We chose not to apply mixed models and/or various univariate regression analyses for each primary and secondary outcome, with time as a continuous variable, as this would lead to a table with a lot of variables with effect sizes, which might be harder to interpret for the reader than comparing means and proportions. Furthermore, it is questionable whether these analyses would lead to significant/interpretable difference. The age effect might not be linear for example. That’s why we applied this dichotomous approach.

Results

Good presentation of results with good use of tables

Thank you for this compliment.

Discussion

I think the message of your discussion and conclusion should be altered, It ambiguously implies that a delay in screening is not associated with inferior outcomes. It should be highlighted that the delay is only 1.5 weeks (17.3 vs. 15.8 weeks). This delay should also be discussed in the context of the wider literature – what is already known about the optimum age for diagnosis and at what point a delay in diagnosis is associated with worsened outcomes. Based on this you should highlight to the reader what new information your research brings to the wider research base. Further discussion on the topics raised in the methods comments is recommended.

This article contains 2 separate approaches

  • We compare to time periods (Covid period vs. period before Covid), assessing delayed screening in the COVID period. In these analyses, there was a mean difference in age at time of screening of 1.5 weeks, and a proportion of 57.6% delayed in the Covid group, vs. 36.7 in the pre-COVID group.
  • We compared outcomes at 1 year for patients with “standard screening” (i.e. <14 weeks) from BOTH periods of time, with patients with “delayed screening” (i.e. >15 weeks) from BOTH periods of team. This lead to a group with delayed screening with a mean of 20.09 weeks and a group with standards screening of 13.09 weeks.

With these approaches, we didn’t find any differences in outcomes at 1 year.

We now have clarified these 2 approaches more in the Methods section (lines 142-144)

Currently, our discussion focusses on the primary outcome (screening during Covid pandemic), we now have elaborated more on the timing of screening in our discussion and what our study adds to the current literature on the topic (lines 342-352). We also mentioned this briefly in our conclusion (lines 403-404).

Round 2

Reviewer 2 Report

Comments and Suggestions for Authors

Authors have addressed all points as requested, thank you